# Knowledge of Primary Care Patients Living in the Urban Areas about Risk Factors of Arterial Hypertension

**DOI:** 10.3390/ijerph20021250

**Published:** 2023-01-10

**Authors:** Tomasz Sobierajski, Stanisław Surma, Monika Romańczyk, Maciej Banach, Suzanne Oparil

**Affiliations:** 1Faculty of Applied Social Sciences and Resocialization, University of Warsaw, 00-927 Warsaw, Poland; 2Faculty of Medical Sciences in Katowice, Medical University of Silesia, 40-752 Katowice, Poland; 3Department of Preventive Cardiology and Lipidology, Medical University of Lodz, 93-338 Lodz, Poland; 4Cardiovascular Research Centre, University of Zielona Gora, 65-417 Zielona Gora, Poland; 5Department of Cardiology and Adult Congenital Heart Diseases, Polish Mother’s Memorial Hospital Research Institute (PMMHRI), 93-338 Lodz, Poland; 6Department of Medicine, School of Medicine, University of Alabama at Brimingham, Brimingham, AL 35294, USA

**Keywords:** cardiovascular risk, education, prevention, public health, society

## Abstract

Arterial hypertension (AH), one of the most common diseases of civilization, is an independent risk factor for cardiovascular morbidity and mortality. This disease is the second, after lipid disorders, the most common cardiovascular risk factor and a significant cause of premature death. In Poland, one in three adults (approximately 11 million people) suffers from it. The aim of our survey was to determine patients’ knowledge of the factors (e.g., age, smoking cigarettes, drinking coffee, shift work) that may influence the development of hypertension. The survey was conducted among 205 adult primary care patients living in urban areas. There was a high correlation between patients’ education and risk factors of AH, such: as excess salt in the diet (*p* = 0.038), smoking electronic cigarettes (*p* = 0.005), moderate alcohol consumption (*p* = 0.028), moderate daily physical activity (*p* = 0.011), female and male sex (*p* = 0.032 and *p* = 0.012), air pollution (*p* < 0.001) and others. In addition, a statistically significant factor shaping patients’ attitudes toward hypertension prevention was the correlation between the respondents’ education and their parents’ prevalence of hypertension (*p* = 0.40). This study increases the knowledge of patients’ awareness of hypertension. It may serve as guidance for primary care providers to pay special attention to environmental interviews with patients and the patient’s family history for the prevention of hypertension incidence.

## 1. Introduction

Arterial hypertension (AH) has been the most common cardiovascular risk factor for years [1]. Its prevalence has doubled since the 1990s from 331 (95% credible interval: 306–359) million women and 317 (292–344) million men in 1990 to 626 (584–668) million women and 652 (604–698) million men in 2019 [2]. An important contributing factor is that AH often does not cause any symptoms, and half of the affected persons are unaware of having hypertension [2]. Hypertension significantly worsens the quality of life and prognosis of affected persons. Patients with untreated AH have an increased risk of stroke, coronary artery disease, myocardial infarction, and death from any cause [3]. Thus, AH is a critical modern medical problem. The most cost-effective method for preventing hypertension is to raise people’s awareness of the risk factors for this condition, both the classic risk factors (overweight and obesity, excess salt in the diet, and sedentary lifestyle, etc.) and the newer non-classical ones (air pollution, environmental noise, smoking, and electronic cigarettes). It is crucial to counter the widespread misinformation associated with various myths about the causes of AH, which distracts attention from the fundamental risk factors for this disorder.

Since family history and lifestyle play significant roles in the prevention and treatment of AH, we examined the level of knowledge of primary care patients about factors that can cause AH. Surveying patients for their knowledge and attitudes about AH is essential because one in three adults in Poland has hypertension [4]. Implementing this type of multidisciplinary research contributes to patients’ knowledge of the disease and can generate procedures in medical practice that reduce the number of people with AH.

## 2. Materials and Methods

### 2.1. Study Design

The survey’s main objective was to examine the level of knowledge of adult patients in primary care centers about hypertension, one of the world’s most prevalent cardiovascular disorders. One of the most effective ways to gain information on a patient’s knowledge about the factors that can cause hypertension is to survey a selected medical facility. Since COVID-19 vaccination was underway for all adult patients when the survey was implemented, people who were ill and used the facility for services other than a vaccination and healthy people who visited the facility for vaccination were eligible for our study. It ensured that the demographic spread of respondents is wide and not limited to older, chronically ill patients, who are the main beneficiaries of primary care services.

The choice of the medical facility was based on several criteria. First, the medical facility should be a primary care facility. Second, it should be a facility operating in a large metropolitan city due to prevalent risk factors. Third, it should be a facility in a part of the city where people from different generations and incomes live. Fourth, neighborhoods should be affected by the gentrification process; new neighborhoods where only families with children live, and neighborhoods where only people with a similar demographic profile live were excluded. We chose a facility in a very demographically diverse neighborhood in Katowice, one of Poland’s largest and most diverse cities. The study used the PAPI (Pen and Paper Interview) technique. Patients who visited the facility were given a survey questionnaire to fill out. If there were questions, the facility’s personnel were informed and trained to address the patient’s concerns. The survey was voluntary and anonymous, so the patients did not provide their names or surnames on the questionnaire or any other sensitive information that could lead to their identification. After completing the questionnaire, the respondents placed the completed questionnaires into a box to which only one of the researchers had access.

The survey was conducted between December 2020 and February 2021.

### 2.2. The Questionnaire

The questionnaire had three parts. The first part collected demographic information, e.g., the patient’s age, gender, education, height, and weight. The second part included, among other questions, questions on whether the respondent has hypertension or another cardiovascular disease and how often they measured their blood pressure. This section also included questions testing a respondent’s knowledge of the normal blood pressure of a healthy person. The third part included an inventory of 18 potential factors that increase the occurrence of hypertension. The authors subjectively arranged the list of factors based on the latest medical knowledge about hypertension. Each factor was assigned a 4-point scale using the Likert scale: totally disagree (TD), rather disagree (RD), rather agree (RA), or totally agree (TA), as well as the alternative answer: do not know (DK). The respondent’s task was to mark on the scale how much he or she agrees or disagrees that a given factor can cause hypertensive disease. All questions were in the form of closed questions with the possibility of one or more answers.

Before starting the study, a pilot study was conducted among randomly selected patients at the medical center where the study was conducted to evaluate the questionnaire. The pilot study revealed that the questions were understandable and clear to the respondents. The pilot trial was not include in the main study trial. 

### 2.3. Statistical Analysis

All statistical analyses were performed using IBM SPSS Statistics 28.0.1.1. A descriptive analysis was carried out to describe the sample, and the results are presented as frequencies and percentages. The chi-square test was used to compare the frequencies. The scales in the questionnaire were validated using Cronbach’s alpha test. Normality was calculated using Shapiro–Wilk tests. We calculated the sample size assuming the margin of error to be no more than 6% at the 95% confidence level. There were 2599 patients in the medical facility at the time of the survey, so the minimum required sample size for reliable statistical analysis was 200 respondents.

### 2.4. Ethical Considerations

This study was not an interventional clinical trial and was not a study that violated the intimate lives of the respondents, nor did it violate their physical or psychical health. The survey did not collect sensitive data from the respondents. For this reason, approval from university ethics committees was not needed to conduct this type of research in Poland.

The methods used in this study comply with the methodological and legal standards required for implementing this type of research in Poland.

All respondents were adults at the time of the survey and gave informed consent to complete the survey. At the same time, the authors made every effort to conduct the study with due diligence and under the latest ethical guidelines for conducting social research.

## 3. Results

### 3.1. Sociodemographic Characteristics

Six out of ten people surveyed had a university education (61.5%, N = 126), one in four people had no physical activity (25.4%, N = 52), and six out of ten people had hypertension (61.5%, N = 126). The largest age group included those 71–80 years old (41.0%, N = 84) (Table 1).

### 3.2. Characteristics of Respondents in Terms of Non-Hypertensive and Hypertensive Patients

Among those who declared they have hypertension, there were more men than women, and three in four people (75.4%) aged 61 or older (*p* < 0.001) were included. People with primary and secondary education were more likely than those with higher education (*p* = 0.005) to have hypertension, which may be because older people, for historical reasons, tend to have less education in general. Those who had hypertension were significantly more likely to do no physical activity (which is also related to the older age in this group of respondents), to smoke cigarettes, and to measure their blood pressure (Table 2) than those who did not have hypertension.

### 3.3. Knowledge of Hypertension and Risk Factors That Cause Hypertension

Most respondents (85.9%) reported that their normal blood pressure value was up to 140/90 mmHg, 7.3% reported that it was up to 150/90 mmHg, 6.3% reported up to 160/90 mmHg, and one person reported 170/90 mmHg. The respondents were presented with a list of 18 factors (e.g., environmental, genetic, lifestyle) that may influence the occurrence of AH.

#### 3.3.1. Factor: The Presence of Hypertension in Parents

One out of every two people surveyed (52.7%, N = 108) reported that their parents had hypertension. In three cases out of ten, the mother was hypertensive (30.2%, N = 62), in one in nine cases, the father was hypertensive (11.7%, N = 24), and in one in ten cases, both parents had hypertension (10.7%, N = 22).

The opinion that the presence of hypertension in parents increases the risk of their children developing hypertension was “totally agreed” with by one in four respondents (26.8%, N = 55), “rather agreed” with by one in three respondents (35.6%, N = 73), “rather disagreed” with by 6.3% (N = 13) of respondents, and “totally disagreed” with by 2% (N = 4) of respondents. Three in ten respondents (29.3%, N = 60) did not know anything about the issue.

The respondents with higher education were more likely to agree with the above view than those with a primary and secondary degree of education (*p* = 0.040) (Table 3).

#### 3.3.2. Factor: Obesity

The view that obesity increases the risk of developing hypertension was “totally agreed” with by one in two respondents (52.2%, N = 107), “rather agreed” with by one in four respondents (26.3%, N = 54), “rather disagreed” with by 4.4% (N = 9) of respondents, “totally disagreed” with by 2.9% (N = 6) of respondents and stated as unknown by one in seven respondents (14.1%, N = 29).

Women were more likely to agree with the above view than men (*p* = 0.014) (Table 4).

#### 3.3.3. Factor: Excess Salt in the Diet

The view that excess salt in the diet increases the risk of developing hypertension was “totally agreed” with by four out of ten respondents (43.9%, N = 90), “rather agreed” with by one in three respondents (31.2%, N = 64), “rather disagreed” with by 2.9% (N = 6) of respondents, and “totally disagreed” with by 3.9% (N = 8) of respondents, and 18.0% (N = 37) had no knowledge about the subject.

The respondents with higher education were more likely to agree with the above view (*p* = 0.038) (Table 5).

#### 3.3.4. Factor: Smoking Cigarettes

The view that daily smoking of traditional cigarettes increases the risk of developing hypertension was “totally agreed” with by one in two respondents (47.8%, N = 98), “rather agreed” with by one in four respondents (23.9%, N = 49), “rather disagreed” with by 2.4% (N = 5) of respondents, “totally disagreed” with by 3.4% (N = 7) of respondents, and unknown by one in five respondents (22.4%, N = 46).

The view that occasional smoking of traditional cigarettes increases the risk of developing hypertension was “totally agreed” with by one in two respondents (12.2%, N = 25), “rather agreed” with by nearly one in four respondents (22.9%, N = 47), “rather disagreed” with by 18% (N = 37) of respondents, “totally disagreed” with by 6.8% (N = 14) of respondents, and unknown by four in ten respondents (40%, N = 82).

The primary and secondary education respondents were more likely to agree with the above view (*p* = 0.019) (Table 6).

The view that smoking electronic cigarettes increases the risk of developing hypertension was “totally agreed” with by one in five respondents (22.0%, N = 45), “rather agreed” with by one in five respondents (21.5%, N = 44), “rather disagreed” with by 7.3% (N = 15) of respondents, and “totally disagreed” with by 4.4% (N = 9) of respondents. Nearly one in two respondents (44.9%, N = 92) has no opinion on the subject.

The respondents with higher education were slightly more likely to agree with the above view (*p* = 0.005) (Table 6).

#### 3.3.5. Factor: Alcohol Consumption

The view that moderate alcohol consumption increases the risk of developing hypertension was “totally agreed” with by 6.3% (N = 13) of respondents, “rather agreed” with by one in five respondents (22.0%, N = 45), “rather disagreed” with by three in ten respondents (29.3%, N = 60), and “totally disagreed” with by 6.8% (N = 14) of respondents. One in three respondents did not have an opinion about the subject (35.6%, N = 73).

The respondents with primary and secondary education were slightly more likely to agree with the above view (*p* = 0.028) (Table 7).

The view that excessive alcohol consumption increases the risk of developing hypertension was “totally agreed” with by one in three respondents (37.6%, N = 77), “rather agreed” with by one in four respondents (26.3%, N = 54), “rather disagreed” with by 8.3% (N = 17) of respondents, and “totally disagreed” with by 3.9% (N = 8) of respondents. One in four respondents had no opinion on the subject (23.9%, N = 49).

#### 3.3.6. Factor: Shift work

The view that shift work increases the risk of developing hypertension was “totally agreed” with by 6.3% (N = 13) of respondents, “rather agreed” with by one in five respondents (19.0%, N = 39), “rather disagreed” with by one in four respondents (19.5%, N = 40), and “totally disagreed” with by one in nine respondents (11.7%, N = 24), and four in ten respondents (43.4%, N = 89) did not have an opinion about the subject.

The respondents who did not suffer from hypertension were more likely to agree with the above view (*p* = 0.037) (Table 8).

#### 3.3.7. Factor: Drinking Coffee and Energy Drinks

The view that drinking 2–3 cups of coffee every day increases the risk of developing hypertension was “totally agreed” with by one in ten respondents (10.2%, N = 21), “rather agreed” with by one in five respondents (21.5%, N = 44), “rather disagreed” with by one in four respondents (26.8%, N = 55), and “totally disagreed” with by one in ten respondents (9.8%, N = 20), and 31.7% (N = 65) did not have an opinion about the subject.

The respondents without hypertension were more likely to agree with the above view (*p* = 0.018) (Table 9).

The view that consumption of energy drinks increases the risk of developing hypertension was “totally agreed” with by 15.1% (N = 31), “rather agreed” with by one in four respondents (23.4%, N = 48), “rather disagreed” with by one in ten respondents (10.2%, N = 21), and “totally disagreed” with by 6.3% (N = 13) of respondents. Nearly one in two respondents did not know about this issue (44.9%, N = 92).

The respondents who did not suffer from hypertension were more likely to agree with the above view (*p* = 0.030) (Table 9).

#### 3.3.8. Factor: Aging

The view that aging increases the risk of developing hypertension was “totally agreed” with by one in seven respondents (14.6%, N = 30), “rather agreed” with by one in three respondents (34.6%, N = 71), “rather disagreed” with by one in ten respondents (10.7%, N = 22), and “totally disagreed” with by 7.8% (N = 16) of respondents, and 32.2% (N = 66) had no opinion on this subject. Among those who suffer from hypertension, one in three (35.7%) had no opinion on this subject.

There was no statistically significant relationship between aging and demographic characteristics of respondents (Table 10).

#### 3.3.9. Factor: Moderate Daily Physical Activity

The view that moderate daily physical activity increases the risk of developing hypertension was “totally agreed” with by 2.4% (N = 5) of respondents, “rather agreed” with by 7.3% (N = 15) of respondents, “rather disagreed” with by one in three respondents (36.1%, N = 74), and “totally disagreed” with by one in four respondents (27.3%, N = 56). One in four respondents did not know about the subject (26.8%, N = 55).

The respondents with primary and secondary education were more likely to agree with the above view (*p* = 0.011) (Table 11).

#### 3.3.10. Factor: Female/Male Sex

The view that being a woman increases the risk of developing hypertension was “totally agreed” with by 4.4% (N = 9), “rather agreed” with by 3.4% (N = 7), “rather disagreed” with by one in four respondents (23.4%, N = 48), and “totally disagreed” with by one in four respondents (24.9%, N =5 8) of those surveyed. Nearly one in two respondents did not know about the subject (43.9%, N = 90).

The respondents with primary and secondary education were more likely to agree with the above view (*p* = 0.032) (Table 12).

The view that being male increases the risk of developing hypertension was “totally agreed” with by 7.3% (N = 15) of respondents, “rather agreed” with by one in eleven respondents (8.8%, N = 18), “rather disagreed” with by one in five respondents (19.0%, N = 39), and “totally disagreed” with by one in five respondents (20.0%, N = 41). Nearly one in two respondents did not know about the issue (44.9%, N = 92).

The respondents with higher education were slightly more likely to agree with the above view (*p* = 0.012) (Table 12).

#### 3.3.11. Factor: Air Pollution (Smog)

The view that air pollution increases the risk of developing hypertension was “totally agreed” with by one in six respondents (15.6%, N = 32), “rather agreed” with by one in three respondents (35.6%, N = 73), “rather disagreed” with by one in nine respondents (11.2%, N = 23), and “totally disagreed” with by 7.8% (N = 16) of respondents, and three in ten respondents (29.8%, N = 61) had no opinion on this subject.

The respondents with primary and secondary education were more likely to agree with the above view (*p* < 0.001) (Table 13).

#### 3.3.12. Factor: Noise

The view that noise increases the risk of developing hypertension was “totally agreed” with by one in ten respondents (10.7%, N = 22), “rather agreed” with by one in four respondents (26.8%, N = 55), “rather disagreed” with by 17.6% (N = 36) of respondents, and “totally disagreed” with by 7.8% (N = 16) of respondents, and one in three respondents (37.1%, N = 76) had no opinion on this subject.

The respondents with primary and secondary education were more likely to agree with the above view (*p* < 0.001) (Table 14).

#### 3.3.13. Factor: Travel to High Mountains, above 2500 Meters above Sea Level

The view that traveling to high mountains increases the risk of developing hypertension was “totally agreed” with by 8.3% (N = 17) of respondents, “rather agreed” with by one in five respondents (22.0%, N = 45), “rather disagreed” with by 18% (N = 37) of respondents, and “totally disagreed” with by one in ten respondents (9.8%, N = 20), and four in ten respondents (42%, N = 86) had no opinion on this subject.

The respondents with primary and secondary education were slightly more likely to agree with the above view (*p* = 0.024) (Table 15).

## 4. Discussion

This exploratory study aimed to learn about patients’ knowledge of factors that may cause hypertension. Based on the results of this study, hypotheses can be formed, which can be tested in in-depth studies on the factors that cause hypertension in humans.

Hypertension is one of the most critical healthcare challenges in Poland and the world. Regardless of the level of development of a country or social group, hypertension is “democratic” and affects many people. It is one of the most common forms of cardiovascular disease [1,2]. Hypertension, classified as an epidemic, limits the physical ability of the sufferer and burdens the healthcare system [5,6,7,8]. Hypertension is caused by many factors that include factors beyond our control (gender, age) and social factors (lifestyle, physical activity, obesity). Blood pressure correlates with body mass index [9,10] and socioeconomic factors (living in a noise and smog-prone area). For this reason, it is essential to determine patients’ knowledge and awareness of the factors that can cause hypertensive disease.

Our study showed several significant statistical relationships between specific factors that can cause hypertension and the respondents’ education level. Education was strongly correlated with several factors, such as excess salt in the diet (*p* = 0.038), occasional smoking of traditional cigarettes (*p* = 0.019), smoking electronic cigarettes (*p* = 0.005), moderate alcohol consumption (*p* = 0.028), moderate daily physical activeness (*p* = 0.011), noise (*p* < 0.001), pollution (*p* < 0.001), traveling to high mountains, above 2500 m above sea level (*p* = 0.024), female sex (*p* = 0.032) and male sex (*p* = 0.012). It is consistent with the results of many other studies, which showed that the level of education not only affects the treatment of hypertension and its control but also leads to a particular lifestyle, which determines or does not determine the onset of hypertension in a person [11,12]. A person’s level of education plays a significant role in disease prevention and the treatment of hypertension, and greater health literacy is strongly correlated with education and can favorably affect a person’s health [13,14,15,16,17].

Our study showed a statistically significant relationship between the educational level of the studied patients and having parents who had hypertension (*p* = 0.040). Family history is a crucial element in studying the causes of AH, which was confirmed by other studies [18,19]. Niiranen et al. studied the risks associated with early-onset hypertension compared to late-onset hypertension among patients. They found that early- but not late-onset hypertension in parents predicted the incidence of hypertension in their offspring [20]. Reaching out to family history is essential in preventing AH incidence in the next generation, not only for genetic inheritance reasons but also for environmental reasons related to lifestyle. Children of parents with AH, during the socialization process, learn specific behaviors and risky points of view on contracting AH, which they often unknowingly replicate in adulthood [21]. 

This study also showed a statistically significant relationship between shift work and the possibility of AH (*p* = 0.037). A Korean study that compared the prevalence of hypertension between day workers and workers who worked shift work (including the night shift) found that as the number of years in shift work increased, the risk of developing AH increased. In workers who had worked shift work for more than 20 years, this risk was significantly higher [22]. Moreover, while there have been papers that did not confirm this relationship [23,24], a 2021 meta-analysis by Gamboa Madeira et al. found that workers who worked night shifts and rotating workers who worked day and night shifts had statistically significant increases in systolic and diastolic blood pressure values. As the authors point out, the effect size was small but significant enough that shift work with night shift and night shift work should be taken as a factor in the occurrence of AH [25].

In recent years, the view on the impact of regular coffee consumption on the risk of hypertension has changed. In a comprehensive review of the literature by Surma and Oparil, it was shown that regular consumption of 2–3 cups of coffee does not affect or reduces the risk of hypertension [26]. We showed that patients with hypertension were significantly more likely to have evidence-based medicine (EBM) knowledge about the impact of regular consumption of 2–3 cups of coffee on the risk of this disease.

Despite many educational campaigns on AH, such as May Measurement Month (MMM), the global campaign run by the International Society of Hypertension, the knowledge of the public and future medical professionals (medical students) about AH is insufficient to inform major improvements in the diagnosis and treatment of AH [27,28,29]. It should be emphasized that insufficient knowledge about AH has been present for many years. A study of Michalska et al. found that efforts should be made to improve knowledge of AH, especially among the rural population, elderly patients, those with a low education level, and young males who had the highest blood pressure [30].

### Limitations of the Study

The main limitation of our survey is that it is not representative of the entire population; therefore, only small- and medium-range hypotheses can be built based on the results. Nevertheless, in carrying out a study of this scope, we took care that the purposive selection was conducted very well. Thus, we obtained data that can be used for further research, including a random, representative sample of the entire population. Another limitation of the study was that it was conducted at a clinic in the city without considering the attitudes of patients who live in the countryside. The demographic category of rural and urban residents is very complicated in Poland due to suburbanization and socio-demographic changes in the countryside. For this reason, to a much greater extent, the lifestyle of a Polish resident is told by his or her education and material status than by the place of residence. Another limitation is the list of factors that influence the occurrence of hypertension. Although the list of factors is exhaustive, we realize the complexity of the process by which the incidence of AH occurs.

## 5. Conclusions

The study showed how knowledge and environmental factors, mainly those related to being raised in a family where the older generation had AH, play a significant role in AH prevention. One of the primary practical implications of our study is that it is imperative to pay attention to the family history of AH in the daily practice of medicine in primary care settings. At the micro-level, this is important for the patient, in whom preventive measures can be put in place early to prevent AH. At the macro-level, it is crucial for public health and advocacy efforts to reduce the growing number of AH patients. Every effort should be made to change this situation soon. In addition, further statistical analysis using more sophisticated statistical methods, such as multiple regression, is recommended to identify relationships between hypertension risk factors and education level, especially among the oldest members of society.

## Figures and Tables

**Table 1 ijerph-20-01250-t001:** Sociodemographic Characteristics (N = 205).

	N (%)
Gender	
Female	106 (51.7)
Male	99 (48.3)
Age (years)	
18–40	20 (9.7)
41–60	43 (21.0)
61–70	36 (17.6)
71–80	84 (41.0)
>80	22 (10.7)
Education	
Primary	8 (3.9)
Secondary	71 (34.6)
Tertiary	126 (61.5)
145–160	46 (22.4)
161–175	112 (54.6)
176 and more	47 (23.0)
*Mean*	*168.5*
Weight (kg)	
Less than 60	38 (18.5)
61–70	48 (23.4)
71–80	55 (26.9)
81–90	50 (24.4)
91 and more	14 (6.8)
*Mean*	*74.7*
Physical activity (daily)	
None	52 (25.4)
About 30 min	80 (39.0)
About 60 min	45 (22.0)
More than 60 min	28 (13.7)
How often do you measure your blood pressure?	
Daily	41 (20.0)
At least once a week	52 (25.4)
At least once a month	73 (35.6)
At least once a year	28 (13.7)
Not at all	11 (5.4)
Do you suffer from any of the following disease?	
Arterial Hypertension	126 (61.5)
Coronary Artery Disease	38 (18.5)
Overweight/Obesity	40 (19.5)
Heart Failure	24 (11.7)
Elevator Cholesterol Levels	52 (25.4)
Myocardial Infraction	18 (8.8)
Diabetes	18 (8.8)
Stroke	14 (6.8)
None of the above	45 (22.0)
Do you smoke cigarettes?	
I do	159 (77.6)
I do not	46 (22.4)
What is your systolic blood pressure? (mm Hg)	
<120	19 (9.2)
120–129	40 (19.5)
130–139	53 (25.9)
140–159	45 (22.0)
160–179	6 (2.9)
>180	0 (0.0)
I do not know	42 (20.5)
What is your diastolic blood pressure? (mmHg)	
<80	49 (23.9)
80–84	61 (29.7)
85–89	20 (9.8)
90–99	30 (14.6)
100–109	3 (1.5)
>110	0 (0.0)
I do not know	42 (20.5)

**Table 2 ijerph-20-01250-t002:** Sociodemographic characteristics of the respondents divided into non-hypertensive and hypertensive subjects (N = 205).

	NonAH *	AH *	*p*-Value
Sex			
Female	48 (45.3)	58 (54.7)	0.040
Male	31 (31.3)	68 (68.7)	
Age			
18–40	19 (95.0)	1 (5.0)	<0.001
41–60	25 (58.1)	18 (41.9)	
61 and more	35 (24.6)	107 (75.4)	
Education			
Primary and Secondary	21 (26.6)	58 (73.4)	0.005
Tertiary	58 (46.0)	68 (54.0)	
Height (cm.)			
145–160	18 (22.8)	28 (22.2)	0.276
161–175	36 (45.6)	76 (60.3)	
176 and more	25 (31.6)	22 (17.5)	
*Mean*	*169*	*168.3*	
Weight (kg.)			
< 60	23 (29.1)	15 (11.9)	0.225
61–70	19 (24.1)	29 (23.0)	
71–80	18 (22.8)	37 (29.4)	
81–90	13 (16.5)	24 (19.0)	
91 and more	6 (7.65)	21 (16.7)	
*Mean*	*70.5*	*77.4*	
Physcial activity daily			
0 min	13 (25.0)	39 (75.0)	0.079
About 30 min	31 (38.8)	49 (61.3)	
About 60 min	22 (48.9)	23 (51.1)	
More than 60 min	13 (46.4)	15 (53.6)	
Do you smoke cigarettes?			
Everyday	11 (26.8)	30 (73.2)	<0.001
At least once a week	13 (25.0)	39 (75.0)	
At least once a month	26 (35.6)	47 (64.4)	
At least once a year	21 (75.0)	7 (25.0)	
I do not measure at all	8 (72.7)	3 (27.3)	
Do you measure your blood pressure?			
I do	12 (26.1)	34 (73.9)	0.049
I do not	67 (42.1)	92 (57.9)	

* NonAH—Non-hypertensive (N = 79), AH—Hypertensive (N = 126).

**Table 3 ijerph-20-01250-t003:** Presence of hypertension in parents as a factor that increases the incidence of hypertension by gender, education, and having hypertension (N = 205).

	Sex			Education			Having AH		
	F	M	*p*-Value	1° 2°	3°	*p*-Value	NonAH	AH	*p*-Value
TA	31 (29.2)	24 (24.2)	0.735	24 (30.4)	31 (24.6)	0.040	18 (22.8)	37 (29.4)	0.109
RA	33 (31.1)	40 (40.4)		23 (29.1)	50 (39.7)		36 (45.6)	37 (29.4)	
RD	7 (6.6)	6 (6.1)		1 (1.3)	12 (9.5)		5 (6.3)	8 (6.3)	
TD	2 (1.9)	2 (2.0)		2 (2.5)	2 (1.6)		0 (0.0)	4 (3.2)	
DK	33 (31.1)	27 (27.3)		29 (36.7)	31 (24.6)		20 (25.3)	40 (31.7)	

Note. Lines: TA—totally agree, RA—rather agree, RD—rather disagree, TD—totally disagree, DK—do not know; Columns: F—females (N = 106), M—males (N = 99); 1° 2°—primary and secondary degree (N = 79), 3°—tertiary degree (N = 126); NonAH—Non-hypertensive (N = 79); AH—Hypertensive (N = 126).

**Table 4 ijerph-20-01250-t004:** Obesity as a factor that increases the incidence of hypertension by gender, education, and having hypertension (N = 205).

	Sex			Education			Having AH		
	F	M	*p*-Value	1° 2°	3°	*p*-Value	NonAH	AH	*p*-Value
TA	63 (59.4)	44 (44.4)	0.014	39 (49.4)	68 (54.0)	0.168	45 (57.0)	62 (49.2)	0.522
RA	25 (23.6)	29 (29.3)		23 (29.1)	31 (24.6)		22 (27.8)	32 (25.4)	
RD	2 (1.9)	7 (7.1)		3 (3.8)	6 (4.8)		2 (2.5)	7 (5.6)	
TD	0 (0.0)	6 (6.1)		5 (6.3)	1 (0.8)		2 (2.5)	4 (3.2)	
DK	16 (15.1)	13 (13.1)		9 (11.4)	20 (15.9)		8 (10.1)	21 (16.7)	

Note. Lines: TA—totally agree, RA—rather agree, RD—rather disagree, TD—totally disagree, DK—do not know; Columns: F—females (N = 106), M—males (N = 99); 1° 2°—primary and secondary degree (N = 79), 3°—tertiary degree (N = 126); NonAH—Non-hypertensive (N = 79), AH—Hypertensive (N = 126).

**Table 5 ijerph-20-01250-t005:** Excess salt in the diet as a factor that increases the incidence of hypertension by gender, education, and having hypertension (N = 205).

	Sex			Education			Having AH		
	F	M	*p*-Value	1° 2°	3°	*p*-Value	NonAH	AH	*p*-Value
TA	50 (47.2)	40 (40.4)	0.388	32 (40.5)	58 (46.0)	0.038	40 (50.6)	50 (39.7)	0.247
RA	33 (31.1)	31 (31.3)		23 (29.1)	41 (32.5)		25 (31.6)	39 (31.0)	
RD	1 (0.9)	5 (5.1)		1 (1.3)	5 (4.0)		1 (1.3)	5 (4.0)	
TD	3 (2.8)	5 (5.1)		7 (8.9)	1 (0.8)		1 (1.3)	7 (5.6)	
DK	19 (17.9)	18 (18.2)		16 (20.3)	21 (16.7)		12 (15.2)	25 (19.8)	

Note. Lines: TA—totally agree, RA—rather agree, RD—rather disagree, TD—totally disagree, DK—do not know; Columns: F—females (N = 106), M—males (N = 99); 1° 2°—primary and secondary degree (N = 79), 3°—tertiary degree (N = 126); NonAH—Non-hypertensive (N = 79), AH—Hypertensive (N = 126).

**Table 6 ijerph-20-01250-t006:** Smoking cigarettes as a factor that increases the incidence of hypertension by gender, education, and having hypertension (N = 205).

	Sex			Education			Having AH		
	F	M	*p*-Value	1° 2°	3°	*p*-Value	NonAH	AH	*p*-Value
	Daily smoking to traditional cigarettes								
TA	48 (43.5)	50 (50.5)	0.685	33 (41.8)	65 (51.6)	0.081	42 (53.2)	56 (44.4)	0.471
RA	26 (24.5)	23 (23.2)		22 (27.8)	27 (21.4)		17 (21.5)	32 (25.4)	
RD	4 (3.8)	1 (1.0)		0 (0.0)	5 (4.0)		1 (1.3)	4 (3.2)	
TD	3 (2.8)	4 (4.0)		5 (6.3)	2 (1.6)		1 (1.3)	6 (4.8)	
DK	25 (23.6)	21 (10.2)		19 (24.1)	27 (21.4)		18 (22.8)	28 (22.2)	
	Occasional smoking of traditional cigarettes, e.g., once a month								
TA	10 (9.4)	15 (15.2)	0.228	12 (15.2)	13 (10.3)	0.019	8 (10.1)	17 (13.5)	0.124
RA	23 (21.7)	24 (24.2)		18 (22.8)	29 (23.0)		22 (27.8)	25 (19.8)	
RD	24 (22.6)	13 (13.1)		8 (10.1)	29 (23.0)		18 (22.80	19 (15.1)	
TD	5 (4.7)	9 (9.1)		10 (12.7)	4 (3.2)		2 (2.5)	12 (9.5)	
DK	44 (41.5)	38 (38.4)		31 (39.2)	51 (40.5)		29 (36.7)	53 (42.1)	
	Smoking electronic cigarettes								
TA	22 (20.8)	23 (23.2)	0.357	16 (20.3)	29 (23.0)	0.005	19 (24.1)	26 (20.6)	0.626
RA	24 (22.6)	20 (20.2)		10 (12.7)	34 (27.0)		20 (25.3)	24 (19.0)	
RD	5 (4.7)	10 (10.1)		6 (7.6)	9 (7.1)		5 (6.3)	10 (7.9)	
TD	3 (2.8)	6 (6.1)		8 (10.1)	1 (0.8)		2 (2.5)	7 (5.6)	
DK	52 (49.1)	40 (40.4)		39 (49.4)	53 (42.1)		33 (41.8)	59 (46.8)	

Note. Lines: TA—totally agree, RA—rather agree, RD—rather disagree, TD—totally disagree, DK—do not know; Columns: F—females (N = 106), M—males (N = 99); 1° 2°—primary and secondary degree (N = 79), 3°—tertiary degree (N = 126); NonAH—Non-hypertensive (N = 79), AH—Hypertensive (N = 126).

**Table 7 ijerph-20-01250-t007:** Alcohol consumption as a factor that increases the incidence of hypertension by gender, education, and having hypertension (N = 205).

	Sex			Education			Having AH		
	F	M	*p*-Value	1° 2°	3°	*p*-Value	NonAH	AH	*p*-Value
	Moderate alcohol consumption								
TA	6 (5.7)	7 (7.1)	0.568	4 (5.1)	9 (7.1)	0.028	3 (3.8)	10 (7.9)	0.285
RA	19 (17.9)	26 (12.7)		23 (29.1)	22 (17.5)		20 (25.3)	25 (19.80	
RD	35 (33.0)	25 (25.3)		16 (20.3)	44 (34.9)		27 (34.2)	33 (26.2)	
TD	7 (6.6)	7 (7.1)		9 (11.4)	5 (4.0)		3 (3.8)	11 (8.7)	
DK	39 (36.8)	34 (34.3)		27 (34.2)	46 (36.5)		26 (32.9)	47 (37.3)	
	Excessive alcohol consumption								
TA	40 (37.7)	37 (37.4)	0.453	33 (41.8)	44 (34.9)	0.233	32 (40.5)	45 (35.7)	0.554
RA	29 (27.4)	25 (25.3)		20 (25.3)	34 (27.0)		24 (30.4)	30 (23.8)	
RD	6 (5.7)	11 (11.1)		9 (11.4)	8 (6.3)		5 (6.3)	12 (9.5)	
TD	6 (5.7)	2 (2.0)		4 (5.1)	4 (3.2)		2 (2.5)	6 (4.8)	
DK	25 (23.6)	24 (24.2)		13 (16.5)	36 (28.6)		16 (20.3)	33 (26.2)	

Note. Lines: TA—totally agree, RA—rather agree, RD—rather disagree, TD—totally disagree, DK—do not know; Columns: F—females (N = 106), M—males (N = 99); 1° 2°—primary and secondary degree (N = 79), 3°—tertiary degree (N = 126); NonAH—Non-hypertensive (N = 79), AH—Hypertensive (N = 126).

**Table 8 ijerph-20-01250-t008:** Shift work as a factor that increases the incidence of hypertension by gender, education, and having hypertension (N = 205).

	Sex			Education			Having AH		
	F	M	*p*-Value	1° 2°	3°	*p*-Value	NonAH	AH	*p*-Value
TA	7 (6.6)	6 (6.1)	0.893	8 (10.1)	5 (4.0)	0.413	5 (6.3)	8 (6.3)	0.037
RA	23 (21.7)	16 (16.2)		12 (15.2)	27 (21.4)		23 (29.1)	16 (12.7)	
RD	20 (18.9)	20 (20.2)		15 (19.0)	25 (19.8)		10 (12.7)	30 (23.80	
TD	12 (11.3)	12 (12.1)		9 (11.4)	15 (11.9)		9 (11.4)	15 (11.9)	
DK	44 (41.5)	45 (45.4)		35 (44.3)	54 (42.9)		32 (40.5)	57 (45.2)	

Note. Lines: TA—totally agree, RA—rather agree, RD—rather disagree, TD—totally disagree, DK—do not know; Columns: F—females (N = 106), M—males (N = 99); 1° 2°—primary and secondary degree (N = 79), 3°—tertiary degree (N = 126); NonAH—Non-hypertensive (N = 79), AH—Hypertensive (N = 126).

**Table 9 ijerph-20-01250-t009:** Drinking coffee and energy drinks as a factor that increases the incidence of hypertension by gender, education, and having hypertension (N = 205).

	Sex			Education			Having AH		
	F	M	*p*-Value	1° 2°	3°	*p*-Value	NonAH	AH	*p*-Value
	Drinking 2–3 cups of coffee every day								
TA	11 (10.4)	10 (10.1)	0.539	12 (15.2)	9 (7.1)	0.124	8 (10.1)	13 (10.3)	0.018
RA	26 (24.5)	18 (18.2)		20 (25.3)	24 (19.0)		26 (32.9)	18 (14.3)	
RD	31 (29.2)	24 (24.2)		20 (15.3)	35 (27.8)		17 (21.5)	38 (30.2)	
TD	9 (8.5)	11 (11.1)		4 (5.1)	16 (12.7)		9 (11.4)	11 (8.7)	
DK	29 (27.4)	36 (36.4)		23 (29.1)	42 (33.3)		19 (24.1)	46 (36.5)	
	Drinking energy drinks								
TA	19 (17.9)	12 (12.1)	0.045	11 (13.9)	20 (15.9)	0.085	18 (22.8)	13 (10.3)	0.030
RA	29 (27.4)	19 (19.2)		19 (24.1)	29 (23.0)		22 (27.8)	26 (20.6)	
RD	11 (10.4)	10 (10.1)		13 (16.5)	8 (6.3)		4 (5.1)	17 (13.5)	
TD	2 (1.9)	11 (11.1)		7 (8.9)	6 (4.8)		4 (5.1)	9 (7.1)	
DK	45 (42.5)	47 (47.5)		29 (36.7)	63 (50.0)		31 (39.2)	61 (48.4)	

Note. Lines: TA—totally agree, RA–rather agree, RD—rather disagree, TD—totally disagree, DK—do not know; Columns: F—females (N = 106), M—males (N = 99); 1° 2°—primary and secondary degree (N = 79), 3°—tertiary degree (N = 126); NonAH—Non-hypertensive (N = 79), AH—Hypertensive (N = 126).

**Table 10 ijerph-20-01250-t010:** Aging as a factor that increases the incidence of hypertension by gender, education, and having hypertension (N = 205).

	Sex			Education			Having AH		
	F	M	*p*-Value	1° 2°	3°	*p*-Value	NonAH	AH	*p*-Value
TA	16 (15.1)	14 (14.1)	0.051	10 (12.7)	20 (15.9)	0.070	14 (17.7)	16 (12.7)	0.324
RA	37 (34.9)	34 (34.3)		22 (27.8)	49 (38.9)		32 (40.5)	39 (31.0)	
RD	15 (14.2)	7 (7.1)		14 (17.7)	8 (6.3)		8 (10.1)	14 (11.1)	
TD	3 (2.8)	13 (13.1)		5 (6.3)	11 (8.7)		4 (5.1)	12 (9.5)	
DK	35 (33.0)	31 (31.3)		28 (35.4)	38 (30.2)		21 (26.6)	45 (35.7)	

Note. Lines: TA—totally agree, RA—rather agree, RD–rather disagree, TD—totally disagree, DK—do not know; Columns: F—females (N = 106), M—males (N = 99); 1° 2°—primary and secondary degree (N = 79), 3°—tertiary degree (N = 126); NonAH—Non-hypertensive (N = 79), AH—Hypertensive (N = 126).

**Table 11 ijerph-20-01250-t011:** Moderate daily physical activity as a factor that increases the incidence of hypertension by gender, education, and having hypertension (N = 205).

	Sex			Education			Having AH		
	F	M	*p*-Value	1° 2°	3°	*p*-Value	NonAH	AH	*p*-Value
TA	0 (0.0)	5 (5.1)	0.100	4 (5.1)	1 (0.8)	0.011	1 (1.13)	4 (3.2)	0.055
RA	7 (6.6)	8 (8.1)		7 (8.9)	8 (6.3)		7 (8.9)	8 (6.3)	
RD	39 (36.8)	35 (35.4)		36 (45.6)	38 (30.2)		25 (31.6)	49 (38.9)	
TD	34 (32.1)	22 (22.2)		13 (16.5)	43 (34.1)		30 (38.0)	26 (20.6)	
DK	26 (24.5)	29 (29.3)		19 (24.1)	36 (28.6)		16 (20.3)	39 (31.0)	

Note. Lines: TA—totally agree, RA–rather agree, RD—rather disagree, TD—totally disagree, DK—do not know; Columns: F—females (N = 106), M—males (N = 99); 1° 2°—primary and secondary degree (N = 79), 3°—tertiary degree (N = 126); NonAH—Non-hypertensive (N = 79), AH—Hypertensive (N = 126).

**Table 12 ijerph-20-01250-t012:** Being a woman as a factor that increases the incidence of hypertension by gender, education, and having hypertension (N = 205).

	Sex			Education			Having AH		
	F	M	*p*-Value	1° 2°	3°	*p*-Value	NonAH	AH	*p*-Value
	Female Sex								
TA	3 (2.8)	6 (6.1)	0.061	5 (6.3)	4 (3.2)	0.032	3 (3.8)	6 (4.8)	0.391
RA	6 (5.7)	1 (1.0)		1 (1.3)	6 (4.8)		5 (6.3)	2 (1.6)	
RD	31 (29.2)	17 (17.2)		26 (32.9)	22 (17.5)		19 (24.1)	29 (23.0)	
TD	24 (22.6)	27 (27.3)		14 (17.7)	37 (29.4)		21 (26.6)	30 (23.8)	
DK	42 (39.6)	48 (48.5)		33 (41.8)	57 (45.2)		31 (39.2)	59 (46.8)	
	Male sex								
TA	6 (5.7)	9 (9.1)	0.326	7 (8.9)	8 (6.3)	0.012	3 (3.8)	12 (9.5)	0.131
RA	12 (11.3)	6 (6.1)		5 (6.3)	13 (10.3)		11 (13.9)	7 (5.6)	
RD	24 (22.6)	15 (15.2)		24 (30.4)	15 (11.9)		13 (16.5)	26 (20.6)	
TD	20 (18.9)	21 (21.2)		11 (13.9)	30 (23.8)		18 (22.8)	23 (18.3)	
DK	44 (41.5)	48 (48.5)		32 (40.5)	60 (47.6)		34 (43.0)	58 (46.0)	

Note. Lines: TA—totally agree, RA—rather agree, RD—rather disagree, TD—totally disagree, DK—do not know; Columns: F—females (N = 106), M—males (N = 99); 1° 2°—primary and secondary degree (N = 79), 3°—tertiary degree (N = 126); NonAH—Non-hypertensive (N = 79), AH—Hypertensive (N = 126).

**Table 13 ijerph-20-01250-t013:** Air pollution (smog) as a factor that increases the incidence of hypertension by gender, education, and having hypertension (N = 205).

	Sex			Education			Having AH		
	F	M	*p*-Value	1° 2°	3°	*p*-Value	NonAH	AH	*p*-Value
TA	18 (17.0)	14 (14.1)	0.100	11 (13.9)	21 (16.7)	<0.001	17 (21.5)	15 (11.9)	0.337
RA	34 (32.1)	39 (39.4)		29 (36.7)	44 (34.9)		28 (35.4)	45 (35.7)	
RD	13 (12.3)	10 (10.1)		17 (21.5)	6 (4.8)		6 (7.6)	17 (13.5)	
TD	4 (3.8)	12 (12.1)		1 (1.3)	15 (11.9)		6 (7.6)	10 (7.9)	
DK	37 (34.9)	24 (24.2)		21 (26.6)	40 (31.7)		22 (27.8)	39 (31.0)	

Note. Lines: TA—totally agree, RA—rather agree, RD—rather disagree, TD—totally disagree, DK—do not know; Columns: F—females (N = 106), M—males (N = 99); 1° 2°—primary and secondary degree (N = 79), 3°—tertiary degree (N = 126); NonAH—Non-hypertensive (N = 79), AH—Hypertensive (N = 126).

**Table 14 ijerph-20-01250-t014:** Noise as a factor that increases the incidence of hypertension by gender, education, and having hypertension (N = 205).

	Sex			Education			Having AH		
	F	M	*p*-Value	1° 2°	3°	*p*-Value	NonAH	AH	*p*-Value
TA	11 (10.4)	11 (11.1)	0.094	10 (12.7)	12 (9.5)	<0.001	10 (12.7)	12 (9.5)	0.668
RA	29 (27.4)	26 (26.3)		26 (32.9)	29 (23.0)		23 (29.1)	32 (25.4)	
RD	20 (18.9)	16 (16.2)		21 (26.6)	15 (11.9)		12 (15.2)	24 (19.0)	
TD	3 (2.8)	13 (13.1)		1 (1.3)	15 (11.9)		4 (5.1)	12 (9.5)	
DK	43 (40.6)	33 (33.3)		21 (26.6)	55 (43.7)		30 (38.0)	46 (36.5)	

Note. Lines: TA—totally agree, RA—rather agree, RD—rather disagree, TD—totally disagree, DK—do not know; Columns: F—females (N = 106), M—males (N = 99); 1° 2°—primary and secondary degree (N = 79), 3°—tertiary degree (N = 126); NonAH—Non-hypertensive (N = 79), AH—Hypertensive (N = 126).

**Table 15 ijerph-20-01250-t015:** Travel to high mountains, above 2500 m above sea level, as a factor that increases the incidence of hypertension by gender, education, and having hypertension (N = 205).

	Sex			Education			Having AH		
	F	M	*p*-Value	1° 2°	3°	*p*-Value	NonAH	AH	*p*-Value
TA	9 (8.5)	8 (8.1)	0.083	10 (12.7)	7 (5.6)	0.024	7 (8.9)	10 (7.9)	0.916
RA	21 (19.8)	24 (24.2)		15 (19.0)	30 (23.8)		15 (19.0)	30 (23.8)	
RD	20 (18.9)	17 (17.2)		21 (26.6)	16 (12.7)		14 (17.7)	23 (18.3)	
TD	5 (4.7)	15 (15.2)		7 (8.9)	13 (10.3)		9 (11.4)	11 (8.7)	
DK	51 (48.1)	35 (35.4)		26 (32.9)	60 (47.6)		34 (43.0)	52 (41.3)	

Note. Lines: TA—totally agree, RA–rather agree, RD—rather disagree, TD—totally disagree, DK—do not know; Columns: F—females (N = 106), M—males (N = 99); 1° 2°—primary and secondary degree (N = 79), 3°—tertiary degree (N = 126); NonAH—Non-hypertensive (N = 79), AH—Hypertensive (N = 126).

## Data Availability

The data presented in this study are available on request from the corresponding author.

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
