# Peer review of "Knowledge of Primary Care Patients Living in the Urban Areas about Risk Factors of Arterial Hypertension"

_ijerph, 2023, doi:10.3390/ijerph20021250_

Round 1

Reviewer 1 Report

69 Inclusion / exclusion criteria lead to a high risk of selection bias. Most importantly patients from rural areas were arbitrarily excluded and it is debatable if nowadays this population has less lifestyle risk factors of HA. 

106 sample size calculation method shows that this study results are only valid for a particular medical center in a specific moment in time. 

It is not specified when and over how long period of time the data was collected. 

108 Sample size is small. For this kind of study much larger samples are both necessary and feasible to make the results and conclusion generalizable.

Moreover, the study suffers low response rate – 8%, that carries further risk of bias.

122 “Women outnumbered men in the survey” This statement is not accurate. Using one-sample (one proportion) z-test, p-value shows the proportion difference is not significantly different from the expected.

126 Analysis of sample characteristics (Tab1) shows that the sample not representative for Polish adult population - it is strongly biased towards elder people and higher education. Younger age groups are summarized in buckets 22 and 20 years wide while older groups in 10 years wide – this is confusing.

402 The fact the study sample is not representative for a broader population is mentioned as a limitation, but this should be elaborated in more details – what sources of bias were identified and how this influences the results. This should be also reflected in the tile mentioning “urban” setting.

84 The questionnaire should be published as an appendix to the paper for better understanding of the methodology.

87-88 repeated sentence

126 Tab1 - No BMI is shown despite height and weight data is available. Also, actual “Overweight/Obesity” classification was not established, and it would be more accurete than declared by answering to the questionnaire item “Do you suffer from any of the following disease? Overweight/Obesity”. 

128 “one in three people (75.4%)” – 75% is more like one in four

129 “People with primary and secondary education were more likely than those with higher education (p=0.005) to have hypertension.”

This claim can be valid only after correcting for the effect of age. Elder people have lower levels of education for historical reasons. 

131 “…hypertension were significantly more likely to do no physical activity” 

also please correct for age

134 Tab2 has formatting errors

is repeated twice

“Do you smoke cigarettes? & Do you measure your blood pressure?” Seem not to correspond with the sub headers (categories) that follow.

152 “Men were slightly more likely to agree with the above view (sum of "totally agree" and "rather agree" responses) than women…”

P-value does not support the claim that “men were more likely to agree”. You can not just look into the values and discard the results of statistical hypothesis testing!

153 “…, those with higher education, and those who were not hypertensive, while one in three (31.7%) hypertensive respondents did not know which factors were associated with hypertension (Table 3a).”

Please re-write this sentence to make it clearer.

165 “Women were more likely to agree with the above view than men and those who do not suffer from AH, while one in six (16.7%) of those with hypertension do not have an opiion about the (Table 3b).”

Please explain the second part – woman are compared to normotensive participants? 

Please re-write this sentence to make it clearer. I do not see how the third part of the sentence “…while…” connects to the previous ones.

178 “Women were slightly more likely to agree with the above view, people with higher education and people who do not suffer from hypertension were significantly more likely to agree, while one in five people (19.8%) who suffer from hypertension did not have an opinion about the subject (Table 3c).”

Just like above - P-value does not support the claim that man vs woman and hypertensive vs normotensive are any different.

Also please re-write this sentence to make it clearer.

Please review the whole paper in this respect. Making claims contradictory to statistical analysis of the data is unacceptable in scientific writing. The paper was not systematically reviewed in this regard any further as this is a systematic error. Also please review the discussion towards any unsupported claims.

Misspellings like “opiion” (167) need to be corrected.

Author Response

Thank you very much for your careful review of our manuscript. All of the comments provided were important and were very carefully considered. Below you will find responses to each comment. Thank you especially for pointing out errors in the Results section.

69 Inclusion / exclusion criteria lead to a high risk of selection bias. Most importantly patients from rural areas were arbitrarily excluded and it is debatable if nowadays this population has less lifestyle risk factors of HA. 

It is a significant point. Thank you. We discussed the issue related to the selection of respondents in detail. Considering Poles' demographics and the associated lifestyles, which translate into health and health-seeking behavior, we can say with a high degree of probability that health in Poland is determined more by education and income than by place of residence. It is also related to the significant structural changes in Poland and the phenomenon of suburbanization, which have underscored that the category of rural residents could be more precise. We referred to this in Limitations of the Study. Line 431.

106 sample size calculation method shows that this study results are only valid for a particular medical center in a specific moment in time. 

The survey was not representative in the sense of random sampling. For this reason, it is impossible to build a wide-range theory based on this survey. However, the deliberate and detailed selection of the study site as a representative of social variation allows for building a theory of small coverage and treating the results of this study as part of a fundamental analysis of patients' AH health-seeking behavior and as a basis for preparing extensive, community-wide surveys. We have written about these limitations in the Discussion. In addition, we have separated the Limitations of the Study section in the Discussion.

It is not specified when and over how long period of time the data was collected. 

Thank you for this comment. It was oversight. We have completed this Line 84.

108 Sample size is small. For this kind of study much larger samples are both necessary and feasible to make the results and conclusion generalizable.

Moreover, the study suffers low response rate – 8%, that carries further risk of bias.

Yes, we agree. We are aware of this. But we also know that the research is nevertheless valuable and points to several significant relationships. We write about this in Limitations of the Study.

122 “Women outnumbered men in the survey” This statement is not accurate. Using one-sample (one proportion) z-test, p-value shows the proportion difference is not significantly different from the expected.

It is an excellent point. Thank you. We have reworded it. Line 123.

126 Analysis of sample characteristics (Tab1) shows that the sample not representative for Polish adult population - it is strongly biased towards elder people and higher education. Younger age groups are summarized in buckets 22 and 20 years wide while older groups in 10 years wide – this is confusing.

As we have already mentioned, the sample of patients surveyed is not representative of the population Corresponds to the representation of patients of primary care clinics. In the title of the paper and the Materials and Methods chapter, we write that the study group is primary care patients, not Poles. In this group, the representation and overrepresentation of the elderly are significant. We did not use intervals with decades in the age category because the percentage of respondents in younger age groups was insignificant.

402 The fact the study sample is not representative for a broader population is mentioned as a limitation, but this should be elaborated in more details – what sources of bias were identified and how this influences the results. This should be also reflected in the tile mentioning “urban” setting.

We have completed this in the Limitations of the Study subsection.

84 The questionnaire should be published as an appendix to the paper for better understanding of the methodology.

We attached the English version of the questionnaire to the letter to Editor.

87-88 repeated sentence

Thank you. We deleted it.

126 Tab1 - No BMI is shown despite height and weight data is available. Also, actual “Overweight/Obesity” classification was not established, and it would be more accurete than declared by answering to the questionnaire item “Do you suffer from any of the following disease? Overweight/Obesity”. 

We agree with this comment. BMI would be a very interesting indication. However, in the process of analyzing the data, we dropped it. Perhaps wrongly. In future studies among AH patients, we will undoubtedly pay attention to it.

128 “one in three people (75.4%)” – 75% is more like one in four

We have corrected it.

129 “People with primary and secondary education were more likely than those with higher education (p=0.005) to have hypertension.”

This claim can be valid only after correcting for the effect of age. Elder people have lower levels of education for historical reasons. 

Very important comment. Thank you. We have added the indicated clarification. Line 138

131 “…hypertension were significantly more likely to do no physical activity” 

also please correct for age

Thank you. We have added a clarification. Line 140.

134 Tab2 has formatting errors

is repeated twice

“Do you smoke cigarettes? & Do you measure your blood pressure?” Seem not to correspond with the sub headers (categories) that follow.

All of Table 2 has been duplicated. We have corrected it.

152 “Men were slightly more likely to agree with the above view (sum of "totally agree" and "rather agree" responses) than women…”

P-value does not support the claim that “men were more likely to agree”. You cannot just look into the values and discard the results of statistical hypothesis testing!

This is our mistake. Thank you for this comment. We have corrected it and reviewed the entire work for this.

153 “…, those with higher education, and those who were not hypertensive, while one in three (31.7%) hypertensive respondents did not know which factors were associated with hypertension (Table 3a).”

Please re-write this sentence to make it clearer.

Thank you. We have corrected it. Line 179.

165 “Women were more likely to agree with the above view than men and those who do not suffer from AH, while one in six (16.7%) of those with hypertension do not have an opiion about the (Table 3b).”

Please explain the second part – woman are compared to normotensive participants? 

Please re-write this sentence to make it clearer. I do not see how the third part of the sentence “…while…” connects to the previous ones.

We have removed this sentence, in line with the errors of relying only on values indicated above. 

178 “Women were slightly more likely to agree with the above view, people with higher education and people who do not suffer from hypertension were significantly more likely to agree, while one in five people (19.8%) who suffer from hypertension did not have an opinion about the subject (Table 3c).”

Just like above - P-value does not support the claim that man vs woman and hypertensive vs normotensive are any different.

Also please re-write this sentence to make it clearer.

We have corrected it.

Please review the whole paper in this respect. Making claims contradictory to statistical analysis of the data is unacceptable in scientific writing. The paper was not systematically reviewed in this regard any further as this is a systematic error. Also please review the discussion towards any unsupported claims.

Thank you very much. We have reviewed and rewritten the entire paper, including the discussion from this angle.

Misspellings like “opiion” (167) need to be corrected.

The entire manuscript was once again read and revised by one of our authors, who is a native speaker in English.

Thank you!

Reviewer 2 Report

The authors have tried to address and important issue about the awareness of hypertension, but there are some major concerns with the study.

1. Why not family or community survey was done when patients come to facility. Was it possible to include other family members in the survey as well.

2. The sample size, 200 respondents seems too less for this kind of epidemiological study.

3. The data and sampling seems highly skewed. For an early effective possible intervention, the younger population should be more taken into account. Here more than 50% are above 70 years. It is quite possible that they don’t depict the current knowledge of hypertension as compared to younger ones. The younger ones >40 years are less than 10%?

4. Line 129 to 131, what is the significance of this sentence. What is the relationship between education and hypertension? Table 2 says more educated have more hypertension, while the subsequent tables show vice versa.  And why education is included in every table?

5. Line 129 to 131, what is the significance of this sentence. What is the relationship between education and hypertension? Table 2 says more educated have more hypertension, while the subsequent tables show the opposite results.  And why education is included in every table?

6. The title seems misleading. The study is mainly focused on educational status and hypertension.

7. Only tables 3f and 3g results are significant. What about the other results

8. The authors haven’t mentioned anything about the questionnaire elements like cronbach’ alpha, reliability index, internal consistency and so on..

Author Response

Thank you very much for your careful review of our manuscript. All the comments provided were important and were very carefully considered. Below you will find responses to each comment.

The authors have tried to address and important issue about the awareness of hypertension, but there are some major concerns with the study.

  1. Why not family or community survey was done when patients come to facility. Was it possible to include other family members in the survey as well.

It is a significant point and a topic that needs to be investigated. Because the study was conducted during the pandemic, expanding it to include family members was challenging. Nonetheless, we plan to conduct a "nested" study examining patients and their families for health-seeking behaviors regarding AH.

  1. The sample size, 200 respondents seems too less for this kind of epidemiological study.

We know that the study group is insignificant. Nor is it representative, as discussed in Limitations of the Study - a subsection we added in the Discussion section. Nevertheless, the results of our study are essential in promoting AH health-seeking behavior, especially among urban residents.

  1. The data and sampling seems highly skewed. For an early effective possible intervention, the younger population should be more taken into account. Here more than 50% are above 70 years. It is quite possible that they don’t depict the current knowledge of hypertension as compared to younger ones. The younger ones >40 years are less than 10%?

We assumed that our study population would be primary care patients. In this population, the predominant group is the elderly, and the numerous minor group is younger people. In the future, using the experience of this survey, we want to survey the population of all Poles with a representative random selection of respondents.

  1. Line 129 to 131, what is the significance of this sentence. What is the relationship between education and hypertension? Table 2 says more educated have more hypertension, while the subsequent tables show vice versa.  And why education is included in every table?

In preparing for the study, based on sociological and demographic knowledge of Polish society, we assumed that knowledge of the factors that cause hypertension might be related to educational level. For this reason, we pay so much attention to education in the analysis.

  1. Line 129 to 131, what is the significance of this sentence. What is the relationship between education and hypertension? Table 2 says more educated have more hypertension, while the subsequent tables show the opposite results.  And why education is included in every table?

In Table 2, we show that 54% of those with tertiary education have hypertension vs. 73.4% of those with secondary and primary education (primary, secondary) have hypertension. Thus, more people with lower education have hypertension. However, we also note that the respondents' age should be considered when analyzing these indications. Line 136-142

  1. The title seems misleading. The study is mainly focused on educational status and hypertension.

Thank you for this comment. We have changed the title.

  1. Only tables 3f and 3g results are significant. What about the other results

Statistical significances are in almost all tables in the manuscript related to the relationship between gender, education, and AH disease. We have made the manuscript more detailed and changed the descriptions next to the tables to make this more apparent.

  1. The authors haven’t mentioned anything about the questionnaire elements like cronbach’ alpha, reliability index, internal consistency and so on..

This is our oversight. We have completed it.

Round 2

Reviewer 1 Report

Thank you for swift and detailed revision of the manuscript. Only minor corrections are still necessary. 

Title – consider adding “in urban setting”

Page one, left margin, Citation – please unify with the final title

149 I suggest reframing the statement like:

People with primary and secondary education were more likely than those with higher education (p = 0.005) to have hypertension, which may be due to the fact that older people, for historical reasons, tend to have lower education. in general.

It is highly recommended to carry out further statistical analysis using more advanced statistical methods like multiple regression to rule out such and similar confounding factors. This includes other correlations with education like air pollution etc.

180 Please clarify if p-values in Table 3a and all the following were calculated based on 5x2 (all categories), 4x2 (DK excluded) or 2x2 (TA+RA vs RD+TD). This is not evident from tables notation, not explained in Methods and inconsistent with parts of the text where “agreed” (w/o totally and rather) is mentioned (presumably meaning TA+RA vs RD+TD). Same 189 “Women were more likely to agree with the above view than men (p = 0.014). (Table 3b)”, and the following tables and passages. It has to be clear every time what the p-value stands for.

Author Response

Thank you very much again for the work you put into making our manuscript the best it can be. It was an invaluable help, for which we are very grateful. Hope we can join scientific "forces" in the future, researching the epidemiology of civilization diseases. 

Title – consider adding “in urban setting”

Page one, left margin, Citation – please unify with the final title

Thank you for this idea. We have clarified that it is about patients from urban areas.

149 I suggest reframing the statement like:

People with primary and secondary education were more likely than those with higher education (p = 0.005) to have hypertension, which may be due to the fact that older people, for historical reasons, tend to have lower education. in general.

It is highly recommended to carry out further statistical analysis using more advanced statistical methods like multiple regression to rule out such and similar confounding factors. This includes other correlations with education like air pollution etc.

Thank you. The indication that there "maybe" be such a relationship is very appropriate. We also decided to include the statement in the Conclusions to encourage others to further research, which we also want to do among urban residents in the near future.

180 Please clarify if p-values in Table 3a and all the following were calculated based on 5x2 (all categories), 4x2 (DK excluded) or 2x2 (TA+RA vs RD+TD). This is not evident from tables notation, not explained in Methods and inconsistent with parts of the text where “agreed” (w/o totally and rather) is mentioned (presumably meaning TA+RA vs RD+TD). Same 189 “Women were more likely to agree with the above view than men (p = 0.014). (Table 3b)”, and the following tables and passages. It has to be clear every time what the p-value stands for.

Thank you for this indication. Indeed, some elements may have needed to be clarified. In the Methods, we have dissected each point of the scale. In addition, we corrected the sentence formation in the descriptions before the tables, as it was very confusing at many points.

Thank you.

Reviewer 2 Report

None

Author Response

Dear Reviewer,

thank you very much for accepting the correction we have made.

All the best,
Authors